# Genome-Wide Identification and Characterization of Bovine Fibroblast Growth Factor (FGF) Gene and Its Expression during Adipocyte Differentiation

**DOI:** 10.3390/ijms24065663

**Published:** 2023-03-16

**Authors:** Hui Sheng, Junxing Zhang, Fen Li, Cuili Pan, Mengli Yang, Yuan Liu, Bei Cai, Lingkai Zhang, Yun Ma

**Affiliations:** Key Laboratory of Ruminant Molecular and Cellular Breeding of Ningxia Hui Autonomous Region, School of Agriculture, Ningxia University, Yinchuan 750021, China

**Keywords:** bovine, FGF gene family, genome-wide identification, adipogenic differentiation, collinear analysis

## Abstract

Fibroblast growth factor (FGF) family genes are a class of polypeptide factors with similar structures that play an important role in regulating cell proliferation and differentiation, nutritional metabolism, and neural activity. In previous studies, the FGF gene has been widely studied and analyzed in many species. However, the systematic study of the FGF gene in cattle has not been reported. In this study, 22 FGF genes distributed on 15 chromosomes were identified in the *Bos taurus* genome and clustered into seven subfamilies according to phylogenetic analysis and conservative domains. Collinear analysis showed that the bovine FGF gene family was homologous to *Bos grunniens*, *Bos indicus*, *Hybrid-Bos taurus*, *Bubalus bubalis,* and *Hybrid-Bos indicus*, and tandem replication and fragment replication were the key driving forces for the expansion of the gene family. Tissue expression profiling showed that bovine FGF genes were commonly expressed in different tissues, with *FGF1*, *FGF5*, *FGF10*, *FGF12*, *FGF16*, *FGF17*, and *FGF20* being highly expressed in adipose tissue. In addition, real-time fluorescence quantitative PCR (qRT-PCR) detection showed that some FGF genes were differentially expressed before and after adipocyte differentiation, indicating their diverse role in the formation of lipid droplets. This study made a comprehensive exploration of the bovine FGF family and laid a foundation for further study on the potential function in the regulation of bovine adipogenic differentiation.

## 1. Introduction

Beef is rich in protein, and its amino acid composition is close to the needs of the human body, so it is deeply loved by consumers. With the improvement in living standards, consumers pay more attention to the taste and flavor of beef products, and these factors are affected by the content and distribution of intramuscular fat (IMF) [1]. At the cellular level, adipocyte proliferation (increased number of adipocytes) and differentiation (adipocyte hypertrophy, increased triglyceride accumulation) are the main ways to increase fat content. It is of great significance to increase the intramuscular fat content and improve beef quality by studying and revealing the molecular regulation mechanism of proliferation and differentiation of bovine adipocytes.

Adipogenesis is a biological process closely coordinated by a series of transcriptional cascades and a large number of transcriptional regulatory factors [2]. At present, the most widely studied regulatory factors of adipogenesis are the CCAAT/enhancer-binding protein (C/EBP) and peroxisome proliferator-activated receptor (PPAR) families, which play an important role in the cascade of adipogenesis [3,4]. In addition, varieties of regulatory factors and signaling pathways are involved in the directional differentiation of pluripotent stem cells into precursor adipocytes, including WNT proteins [5], bone morphogenetic protein (BMP), and members of the fibroblast growth factor (FGF) proteins [6,7]. Here, the main purpose of this study is to explore the regulatory role of the FGF gene family in the process of adipocyte differentiation.

The FGF family consists of 22 members that not only closely relate to embryonic development, tissue regeneration, angiogenesis, metabolic activity, and neurological function but also play an important role in cell proliferation, differentiation, migration, apoptosis, and chemotaxis [8,9]. Mitochondrial brown fat uncoupling protein-1 (UCP1) plays a key role in regulating the energy balance of brown adipose tissue (BAT). It has been found that FGF6 and FGF9 can regulate energy metabolism by inducing the expression of UCP1 in adipocytes and preadipocytes [10]. FGF2 not only acts on muscle growth but also promotes fat and angiogenesis [11,12]. FGF10 can not only stimulate the proliferation of preadipocytes through the Ras/MAPK pathway but also promote the expression of retinoblastoma protein (pRb), and the complex of pRb and C/EBPα can induce adipogenesis [13]. In rodent models with obesity and type 2 diabetes, FGF21 has the effect of reducing blood sugar and lipidemia and can increase energy consumption leading to weight loss [14].

Although the functional studies of some FGF family members have been reported in many species, their expression patterns and regulatory mechanisms during adipogenic differentiation of bovine adipocytes have not been systematically studied and elucidated. Therefore, in this study, the characteristics and function of the FGF gene family members are analyzed. Furthermore, we detect the expression profile of FGF family members during the adipogenic differentiation in cattle. Our results lay a foundation for further exploring the molecular mechanism of the FGF genes on bovine adipogenesis.

## 2. Results

### 2.1. Identification of Members of the Bovine FGF Family

In this study, 49 verified FGF protein sequences of humans (*Homo sapiens*, 22), mice (*Mus musculus*, 22), and cattle (*Bos taurus*, 5) were used to identify members of the FGF family. Through the HMM analysis and BLASTP alignment of these 49 protein sequences, 22 non-redundant FGF protein sequences were identified in cattle, including FGF1–FGF14 and FGF16–FGF23 (Table 1). Meanwhile, the corresponding FGF family proteins (Appendix A) were identified in *Bos indicus* (19), *Hybrid-Bos taurus* (21), *Hybrid-Bos indicus* (20), *Bos grunniens* (20), *Bubalus bubalis* (22), *Bos mutus* (16), and *Bison bison bison* (18). All FGF protein sequences can be seen in Appendix A.

Isoelectric point (PI), molecular weight (Mw), and the number and sequence of amino acids (AA) are shown in Appendix A. The results showed that the amino acid sequences of 22 bovine FGF proteins ranged from 155 (FGF1) to 270 (FGF5), while the molecular weight (Mw) ranged from 17249.83 to 29640.87 Da, which was consistent with the corresponding protein length. FGF1 and FGF21 showed acidity with PIs of 6.51 and 6.08, respectively. FGF9 was neutral (7.06), and all other protein members were basic, with PIs between 7.7 and 11.48. All the 22 FGF proteins of bovine contained FGF/FGF superfamily conserved domain (Appendix A).

### 2.2. Structural Characteristics of Members of the Bovine FGF Family

In this study, the phylogenetic relationships of bovine FGF family members were analyzed to predict their conserved motifs and gene structure (Figure 1). The results showed that the FGF family members of cattle were mainly clustered into eight subfamilies according to the different evolutionary branches. The FGF1 subfamily consisted of FGF1 and FGF2, the FGF4 subfamily consisted of FGF4-6, the FGF7 subfamily consisted of FGF7, FGF10, and FGF22, the FGF8 subfamily consisted of FGF8, FGF17, and FGF18, the FGF9 subfamily consisted of FGF9, FGF16, and FGF20, the FGF19 subfamily consisted of FGF19, FGF21 and FGF23, FGF11-14 was from a subfamily, and FGF3 was divided into a separate subfamily. All FGF family proteins contained motif 1 and motif 2, which consisted of 21 and 20 amino acids, respectively (Appendix A). The FGF8 subfamily had the same six motifs, FGF1-4 and FGF6 had the same four motifs, FGF11-14 had the same six motifs, and FGF9, FGF16, and FGF20 had the same five motifs. The coding sequence (CDS), untranslated region (UTR), and intron of FGF family members were different. The number of CDS varied from 2 to 13, and the position and length of 3′ UTR and 5′ UTR were also different, but the members of the FGF family in the same evolutionary branch showed similar conservative patterns and gene structures.

### 2.3. Phylogenetic Analysis of Bovine FGF Protein

On the basis of exploring the evolutionary relationship of the bovine FGF gene, we constructed a phylogenetic tree based on a total of 202 FGF members, including human, mouse, and eight bovine subfamily species (Figure 2). Phylogenetic analysis showed that FGF family proteins were mainly clustered into eight groups, and the number of genes in each group was different. Group IV was the largest with 37 genes, followed by Group III with 30, both Groups I and IV with 28, both Groups II and VII with 26, and Group VIII with 19 and Group VI with 8.

### 2.4. Chromosome Distribution and Collinearity Analysis of FGF Gene

We analyzed the location of FGF family members on the chromosomes of six bovine subfamily species, and the 22 FGF genes in cattle were unevenly distributed on 15 chromosomes (Figure 3). Compared to *Bos taurus*, *Bos indicus* (*FGF3*, *FGF5*, and *FGF17*), *Hybrid-Bos taurus* (*FGF13* and *FGF16*), *Hybrid-Bos indicus* (*FGF13*), and *Bos grunniens* (*FGF17* and *FGF19*) are missing several FGF genes. Meanwhile, the order of *FGF1* and *FGF22* of *Bos taurus* (Chr7) on chromosomes was opposite to that of *Hybrid-Bos taurus* (Chr7), *Hybrid-Bos indicus* (Chr7), and *Bubalus bubalis* (Chr9). In addition, the sequence and position of three tandem genes (*FGF19*, *FGF4*, and *FGF3*) located on *Bos taurus* chromosome 29 changed on *Hybrid-Bos taurus*, *Hybrid-Bos indicus*, and *Bos grunniens* chromosomes. In addition, we found two pairs of tandem repeat genes on the chromosomes of *Bos taurus*. *FGF3*, *FGF4*, and *FGF19* were located on chromosome 29, only 35 and 63 kb apart, respectively. On chromosome 5, *FGF6* and *FGF23* were also located within 50 kb of each other. Meanwhile, eight pairs of fragmented repeat genes were also found (Figure 4). These gene replication events may be one of the drivers of FGF gene evolution.

### 2.5. Collinear Analysis of FGF Gene in Several Bovine Subfamily Species

To further explore the phylogenetic mechanisms of the FGF gene, we studied homology in six bovine subfamily species. The results of the collinear analysis showed that there were multiple collinear gene pairs between *Bos taurus* and *Bos indicus* (31691), *Hybrid-Bos taurus* (34495), *Hybrid-Bos indicus* (33570), *Bos grunnines* (32378), and *Bubalus bubalis* (33327) (Figure 5). There was a one-to-one correspondence between *Bos taurus* chromosomes (2N = 60) and *Hybrid-Bos indicus*, *Bos grunnines*, *Hybrid-Bos taurus*, and *Bos indicus*, and there was also great homology between *Bos taurus* chromosomes and *Bubalus bubalis* chromosomes (2N = 50), indicating that these collinear gene pairs are relatively conservative in the evolution of bovine species (Table 2).

### 2.6. Expression Analysis of FGF Gene in Different Tissues

The expression patterns of genes can provide an important reference for studying their function, so we explored the expression patterns of FGF gene family members in eight tissue types (heart, liver, spleen, lung, kidney, muscle, adipose, rumen) of cattle. (Figure 6, Appendix A). The results showed that the expression of *FGF1*, *FGF5*, *FGF10*, *FGF12*, *FGF16*, *FGF17,* and *FGF20* was the highest in adipose tissue, while the expression of *FGF4*, *FGF7*, *FGF8*, *FGF11*, *FGF14*, *FGF18*, *FGF19*, *FGF21*, *FGF22,* and *FGF23* in adipose tissue was lower than that in lung tissue but higher than that in other tissues. In addition, all the members of the FGF gene family were generally expressed in various tissues, indicating that they may play a wide range of roles in life activities.

### 2.7. Expression Analysis of FGF Gene during Differentiation of Bovine Adipocytes

This study explored the expression pattern of the FGF gene family using bovine subcutaneous adipocytes. The results of Oil Red O staining showed that the number of lipid droplets formed by adipocytes on the 10th day was significantly higher than that of uninduced adipocytes. The results of qRT-PCR detection showed that the expression levels of adipogenic marker genes *FABP4* and *PPARγ* increased significantly after cell induction, indicating that the induced differentiation model of bovine adipocytes was successfully established (Figure 7, Appendix A). Then, the expression pattern of the FGF gene was detected by cell model, and it was found that except for *FGF4*, *FGF13*, *FGF16*, *FGF21,* and *FGF22*, the FGF family genes had relatively high expression levels in bovine adipocytes. In addition, with the increase in induction days, the expression levels of *FGF1*, *FGF2*, *FGF3*, *FGF10*, *FGF11,* and *FGF18* increased significantly. The expression levels of *FGF5*, *FGF10,* and *FGF20* were highest on the second day of differentiation and then decreased rapidly. The expression of *FGF14* was the highest on the 4th day of differentiation and then decreased to the lowest on the 10th day of differentiation. The expression levels of *FGF12* did not change significantly on the 2nd to 8th day of differentiation but decreased significantly on the 10th day of differentiation. The expression level of other FGF family members decreased significantly with the increase in adipocyte differentiation time (Figure 8, Appendix A).

## 3. Discussion

Beef is one of the most important meat products in daily life, and the content of IMF directly affects its taste and flavor, so it is of great significance to explore the molecular mechanism affecting IMF deposition. In recent years, with the completion of whole genome sequencing of animals and plants, a large number of studies on gene families have been reported. Fibroblast growth factor (FGF) transduces signals through fibroblast growth factor receptor (FGFR) tyrosine kinase, which mainly regulates the development and morphogenesis of many tissues by paracrine or autocrine actions [15,16,17]. Considering the potential role of FGF family members and the fact that only a small number of FGF family members have been reported in other species [17,18], we believe that it is very important to identify and analyze the FGF gene family in cattle.

### 3.1. Identification and Phylogenetic Analysis of Bovine FGF Family Proteins

In this study, we used the 49 identified human, mouse, and bovine FGF protein sequences as references to retrieve FGF genes in the *Bos taurus* (22), *Bos indicus* (19), *Hybrid-Bos taurus* (20), *Hybrid-Bos indicus* (21), *Bos grunniens* (20), *Bubalus bubalis* (22), *Bos mutus* (16), and *Bison bison bison* (18) genomes based on sequence similarity and conserved structural domains. The difference in the number of FGF gene family members may be related to the genome size and ploidy level [19]. Previous studies have found that *Caenorhabditis elegans* has only two FGF genes, whereas 22, 22, 27, and 35 FGF genes were identified in the genomes of human, mouse, zebrafish, and common carp, respectively, indicating that the massive expansion of FGF gene family members occurred during the evolution of primitive metazoans into vertebrates and aquatic organisms [17,18,19,20,21]. In addition, a new FGF gene, *FGF24*, has been identified in zebrafish, but direct homologs of *FGF24* have not been identified in humans, mice, or bovids, and it is speculated that it may have been lost during the evolution of these animals [17,22]. The phylogenetic study showed that the FGF genes identified in humans, mice, and eight species of the Bovine subfamily were clustered into eight main branches (Figure 2), and the FGF members with close evolutionary distance would gather together. For example, *Bos taurus FGF1*, *FGF13*, and *FGF23* clustered first with *Bos indicus* and then with FGF genes from other species. Studies have shown that the FGF family has experienced at least two major extensions: The first expansion increased the number of FGFs from one or more primitive FGF genes to eight primitive FGF genes, forming the prototype of eight subfamilies. The second amplification occurred in the process of allelic evolution, which was mainly caused by genome replication [23]. In this study, a phylogenetic tree was constructed using 22 bovine fibroblast growth factor protein sequences, and it was found that they could be clustered into eight subfamilies (FGF1, FGF3, FGF4, FGF7, FGF8, FGF9, FGF11, and FGF19 subfamilies), while the 22 FGF genes in the human and mouse genomes were clustered into seven subfamilies, with FGF3 identified as a member of the FGF7 subfamily [17,21]. In all the current studies, FGF3 genes in vertebrates are always divided into the FGF4 subfamily or FGF7 subfamily, but in fact, the clustering classification of FGF3 is still controversial [17,23,24]. Oulion proposed a new evolutionary scenario for FGF genes based on the results obtained by studying gene content, phylogenetic distribution, and the conservation of commonalities between amphioxus and vertebrates, namely that FGF3 forms a new subfamily, which is consistent with the results of the present study [25]. This evolutionary scheme is demonstrated for the first time in this study based on the results of analyzing the phylogenetic relationships of bovine FGF family members, and it contributes to reconciling different evolutionary hypotheses proposed in previous studies.

### 3.2. Analysis of Physicochemical Properties and Structural Characteristics of the Bovine FGF Protein

Molecular weight and isoelectric point play an important role in determining molecular and biochemical functions [26]. We studied the size and isoelectric point of bovine FGF protein and found that, except for FGF1, FGF9, and FGF21, the isoelectric point of most FGF proteins was more than seven, indicating that there was a high proportion of basic amino acids. In order to gain insight into the structural diversity of bovine FGF proteins, the intron-exon organization was analyzed (Figure 1). Some similar FGF gene pairs showed different intron/exon arrangements, which indicates that the bovine FGF gene may have a more complex gene structure evolution. We identified 10 conserved motifs in bovine FGF proteins, of which motif 1 and motif 2 were present in almost all FGF proteins (Appendix A). There were some differences in the arrangement of conserved motifs among members of the FGF family, but the members of the same subfamily were composed of similar motifs, which indicates that their structures and biological functions are similar. These results confirmed the characteristics of the bovine FGF protein family and laid a foundation for further study of the function of the FGF gene.

### 3.3. Chromosome Distribution, Replication, and Collinearity Analysis of Bovine FGF Protein

The chromosome map of the bovine FGF gene showed that 22 FGF genes were unevenly distributed on 15 chromosomes (Figure 3). The number of genes on each chromosome varied from one to three, including two genes on chromosomes 5, 7, 12, 20, and X, three genes on chromosome 29, and only one gene on most other chromosomes. Gene replication (tandem replication and fragment replication) and transposable events are the main driving forces leading to the complexity of eukaryotic genomes and the expansion of family members [27]. In the study of the human FGF gene family, it was found that *FGF3*, *FGF4*, and *FGF19* were located on chromosome 11, and the distances were 40 and 10 kb, respectively. *FGF6* and *FGF23* were within 55 kb on chromosome 12. In the mouse study, it was found that *FGF3*, *FGF4*, and *FGF19* were located in the 80 kb range of chromosome 7, and *FGF6* and *FGF23* were closely linked on chromosome 6 [28]. Like humans and mice, we identified *FGF3*, *FGF4*, and *FGF19* on bovine chromosomes within the range of 35 to 63 kb on chromosome 29, while *FGF6* and *FGF23* were also located within 50 kb on chromosome 5 (Figure 3). These gene positions suggest that the FGF gene family arose through the duplication and translocation of genes and chromosomes during evolution, which would have contributed to the diversification of gene functions [28]. To investigate the evolutionary relationships of the FGF genes, we performed genomic collinearity analysis on cattle and five other bovine subfamily species and found multiple collinearity gene pairs, indicating that the FGF genes are highly conserved.

### 3.4. FGF Gene Affects Adipocyte Differentiation

In order to understand the expression of FGF family members, the expression profiles of the FGF gene in eight tissue types of cattle were analyzed. The results showed that FGF family members were expressed in all these tissues, with *FGF1*, *FGF5*, *FGF10*, *FGF12*, *FGF16*, *FGF17*, and *FGF20* being highly expressed in adipose tissue. Studies have shown that *FGF1* can promote the differentiation of human preadipocytes into mature adipocytes by regulating the dependent network of BMP protein and activin membrane binding inhibitor (BAMBI)/PPARγ [29]. *FGF2* can activate PI3K/AKT signal pathway and promote the proliferation and adipogenic differentiation of adipose stem cells [30]. *FGF10* can promote adipogenic differentiation of goat intramuscular preadipocytes [31]. This is consistent with our qRT-PCR results, the expression levels of *FGF1*, *FGF2,* and *FGF10* are significantly increased in induced adipocytes, and other FGF family members have different expression levels. Fibroblast growth factor receptors (FGFRs) are tyrosine kinase receptors (TRKs) that include four genes, *FGFR1*, *FGFR2*, *FGFR3*, and *FGFR4* [32]. Four FGFR genes in vertebrates produce seven FGFR proteins with different ligand binding specificity (FGFRs 1b, 1c, 2b, 2c, 3b, 3c, and 4) according to the difference of immunoglobulin-like domain III, and the biological function of typical FGFs is mediated by the interaction with FGFRs [33,34,35]. Therefore, we performed an expression analysis of FGFRs, which showed that *FGFR2* and *FGFR4* were most highly expressed in adipose tissue and *FGFR1* and *FGFR3* were less expressed in adipose tissue than in lung and kidney tissue but higher than in other tissues (Figure 9A, Appendix A). Previous studies on the binding specificity of FGFs-FGFRs found that members of the FGF7 subfamily were able to strongly activate FGFR2b, members of the FGF8 and FGF9 subfamilies showed high relative activity toward FGFR3c, members of the FGF19 subfamily showed consistent activity toward FGFR1c, 2c, 3c and FGFR4, and members of the FGF4 subfamily specifically activated the c receptor splice form [34,35]. Our analysis of the expression of FGFRs during adipocyte lipogenic differentiation also showed that the expression trends of FGFRs were similar to the expression trends of their specifically bound FGF subfamily members, and the findings strongly suggest that FGF–FGFR signaling plays an important role in adipose tissue development and adipocyte lipogenic differentiation (Figure 6, Figure 8 and Figure 9, Appendix A).

The interaction between proteins can reveal their regulatory relationship, which helps us to understand the potential function of these proteins. We used Cytoscape’s Agilent plug-in to mine the literature about FGF family members and their interaction genes to build a complete interaction network (Figure 10) [36]. For example, the FGF signal can regulate the metabolism of endothelial cells through MYC-dependent HK2 expression, which in turn affects the development of blood vessels and lymphatic vessels [37]. Knockout of the *FGF21* gene in liver tissue can activate glucose-6-phosphatase and phosphoenolpyruvate carboxykinase through STAT3/SOCS3 pathway, thus increasing gluconeogenesis and glycogen decomposition, resulting in the aggravation of liver insulin resistance [38]. In addition, studies correlating single nucleotide polymorphisms (SNPs) in the 3′ untranslated region (UTR) of the *FGF21* gene with metabolic syndrome, obesity, and diabetes showed that genetic variants in the 3′ UTR region of the *FGF21* gene were associated with obesity and not with metabolic syndrome and diabetes [39]. The opening of Piezo1 ion channels in mature adipocytes leads to the release of FGF1, which activates FGFR1 and induces precursor adipocyte differentiation [40]. In general, these results showed that the FGF gene family plays a role in regulating vascular development, metabolism, and adipose differentiation by interacting with other genes.

## 4. Materials and Methods

### 4.1. Identification and Phylogenetic Analysis of FGF Gene

We downloaded the genome files and annotation information of related species from the Ensembl database (https://asia.ensembl.org/info/about/species.html, accessed on 11 May 2022) and NCBI database (https://www.ncbi.nlm.nih.gov/genome/?term=BOS, accessed on 13 May 2022), respectively. The hidden Markov model (HMM) of the FGF gene (PF0048) was downloaded from the Pfam database (http://pfam-legacy.xfam.org/, accessed on 25 May 2022), and the HMMER 3.0 software (version 3.0) was used to build a multiple comparison model based on structural domain similarity to retrieve possible FGF proteins according to default parameters [41]. Meanwhile, according to the same template protein sequence, the possible FGF protein was obtained by Protein Basic Local Alignment Search Tool (BLASTP) analysis [42]. Then, the final protein sequences of FGF were obtained by manual examination of the two analysis results, and these protein sequences were submitted to NCBI CD-Search (https://www.ncbi.nlm.nih.gov/Structure/bwrpsb/bwrpsb.cgi, accessed on 5 June 2022) to determine the conserved protein domain [43]. Basic information, such as PI and Mw of genes, is predicted by ExPASy website (https://web.expasy.org/compute_pi/, accessed on 16 June 2022) [44]. The amino acid sequence alignment of the FGF gene was completed by ClustalW software (version 2.1). The results were analyzed by MEGA software (version 7.0.26), and the phylogenetic tree was constructed by using default parameters and setting 1000 repeats [45,46]. The evolutionary tree was adjusted and embellished using Figtree software (version 1.4) [47].

### 4.2. Conservative Motif and Gene Structure Analysis

Motif analysis of the amino acid sequence of FGF through the MEME database (https://meme-suite.org/meme/tools/meme, accessed on 2 July 2022) was conducted. In the parameter setting, the maximum number of motifs was 10, the optimal width was 6–50 amino acids, and the motifs with e values less than 1 × 10^−10^ were retained to identify the conservative motifs in these sequences [48]. The structure of the FGF gene was analyzed using TBtools software (version 1.108), and the structure of the gene was located through CDS and genome sequencing [49].

### 4.3. Chromosome Distribution, Gene Replication, and Collinearity Analysis

Using the genome annotation information obtained, the FGF genes of several bovine subfamily species were mapped to the corresponding chromosomes. MCScanX tool was used to analyze the replication events of the bovine FGF gene, and the collinearity analysis of homologous genes between cattle and five other bovine subfamily species was conducted [50]. All the above results were visualized using TBTools [49].

### 4.4. Culture and Induced Differentiation of Bovine Primary Adipocytes

The subcutaneous adipocytes of cattle were provided by the Key Laboratory of Ruminant Molecular Cell Breeding of Ningxia University. Adipocytes stored in liquid nitrogen were resuscitated and inoculated in culture dishes to grow to about 80%, the differentiation medium was changed to induce differentiation of the cells, and after 2 days of induction, the maintenance medium was changed to continue the culture. The adipocytes that were not induced and induced for 10 days were stained with Oil Red O and photographed and preserved under a microscope. The specific content is carried out with reference to the method [51,52].

### 4.5. RNA Extraction and qRT-PCR

Tissue samples of the heart, liver, spleen, lung, kidney, muscle, adipose, and rumen of cattle were provided by the Key Laboratory of Ruminant Molecular Cell Breeding of Ningxia University. The total RNA of cultured cells was extracted with TRIZOL reagent (American Invitgen), and the purity, concentration, and integrity of RNA were detected by ultraviolet spectrophotometer and 1.0% agarose gel electrophoresis. The first strand cDNA was prepared using a cDNA synthesis kit (Takara, China), and the gene expression level was detected by real-time fluorescence quantitative PCR reaction (qRT-PCR). Primer information is included in Appendix A.

### 4.6. Statistical Analysis

In the qRT-PCR experiment, the mRNA of GAPDH was used as the endogenous control at the basic level, and the relative gene expression level was measured using the 2^−ΔΔCt^ method [53,54]. Visualization of statistical results was performed using GraphPad Prism software (version 7.0).

## 5. Conclusions

Through genome-wide analysis, we identified a total of 158 FGF genes in the Bovidae. The molecular characteristics, gene structure, chromosome location, conserved motif, and evolutionary relationship of FGF genes were analyzed comprehensively. There was a good collinearity of the FGF gene between bovine and other species of bovine subfamily. Expression analysis and functional prediction indicated that FGF genes exhibited tissue-specific expression patterns and that they played an important role in regulating adipocyte differentiation. This study enriched the understanding of the FGF family and laid a foundation for further study of the molecular mechanism of FGF genes in the adipogenic differentiation and development of adipose tissue of cattle.

## Figures and Tables

**Figure 1 ijms-24-05663-f001:**
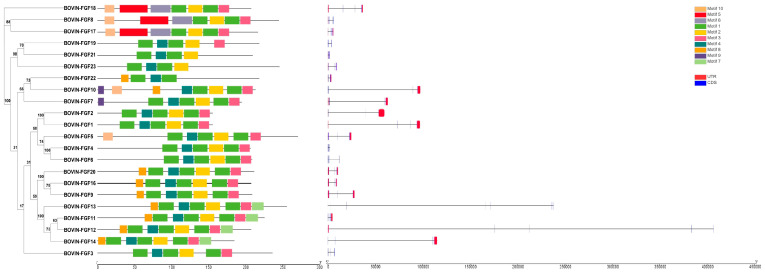
Phylogenetic analysis (**left**), motif analysis (**middle**), and gene structure analysis (**right**) of bovine FGF protein.

**Figure 2 ijms-24-05663-f002:**
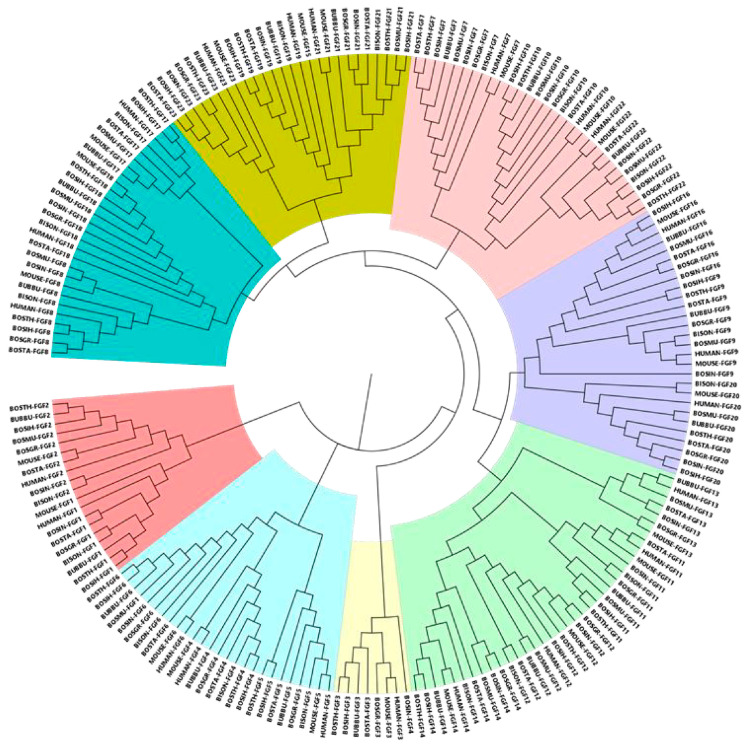
Phylogenetic neighbor-joining (NJ) tree of FGF family members of different species. FGF proteins were divided into eight clusters, which were distinguished by different colors.

**Figure 3 ijms-24-05663-f003:**
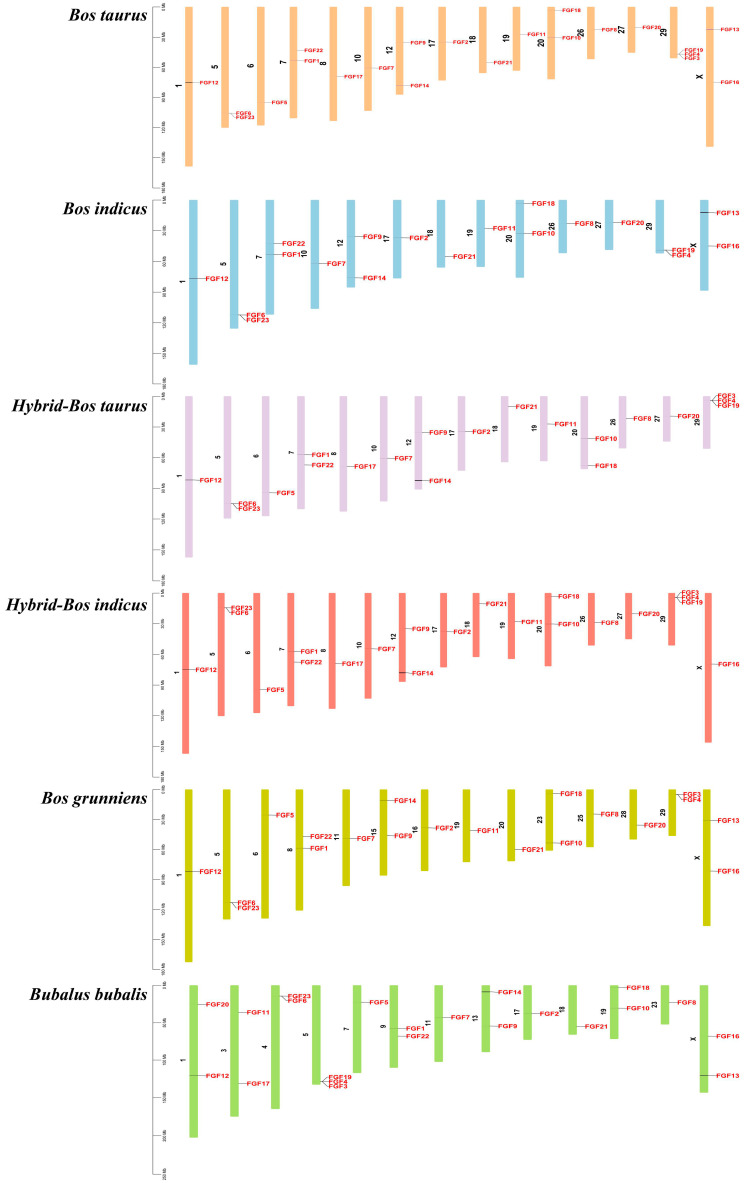
The chromosomal distribution of FGFs gene.

**Figure 4 ijms-24-05663-f004:**
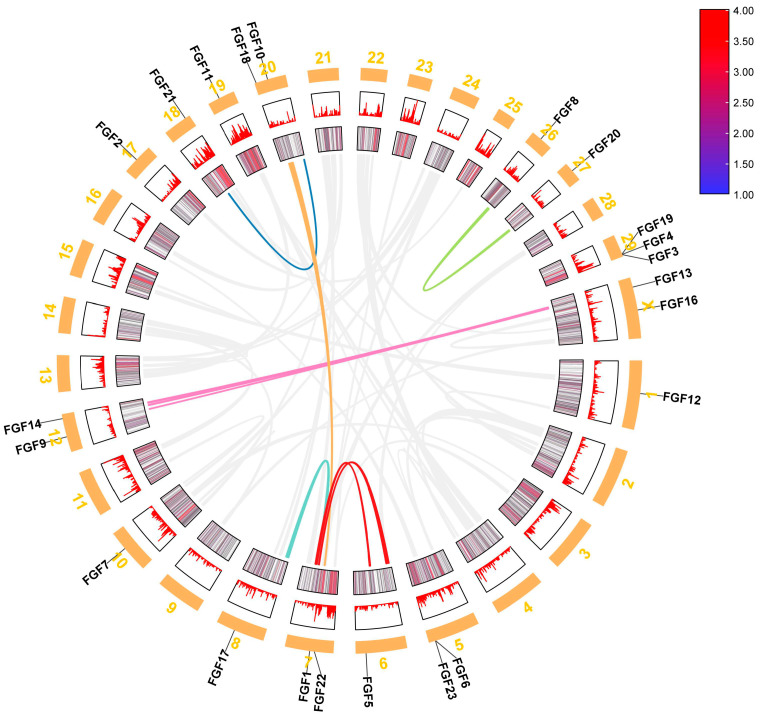
Repetitive gene analysis of FGF gene in *Bos taurus* genome.

**Figure 5 ijms-24-05663-f005:**
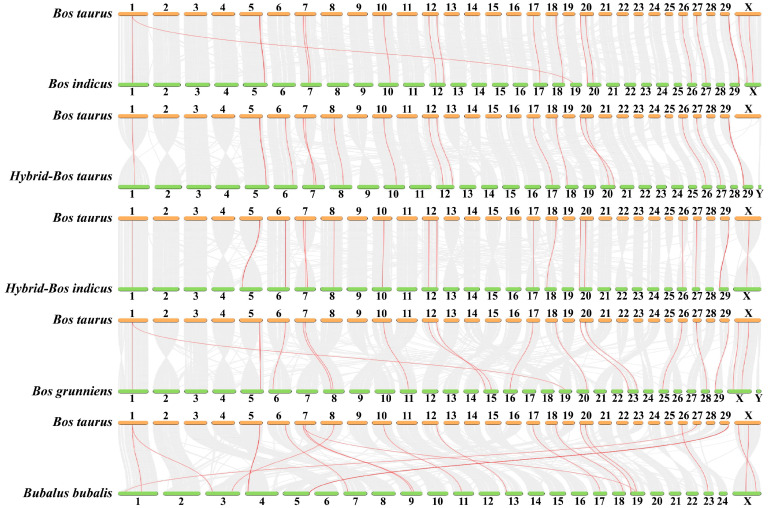
Collinear analysis of FGF gene.

**Figure 6 ijms-24-05663-f006:**
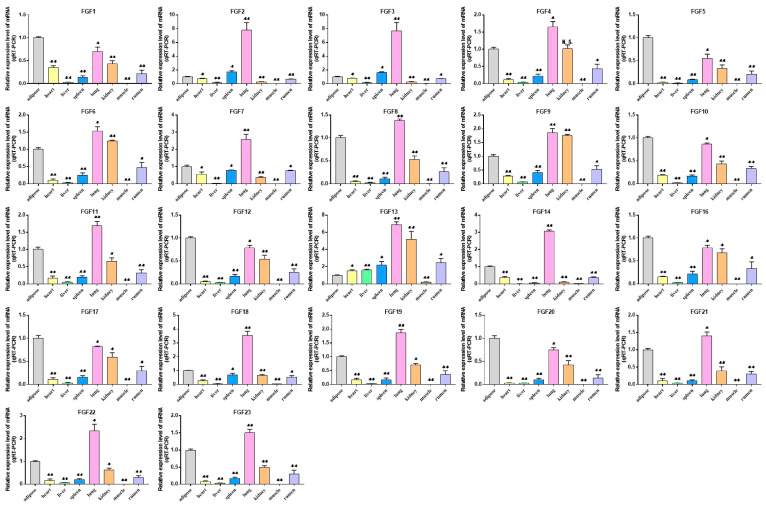
Analysis of the expression of FGF family members in different tissues of cattle. Compared with the control, “*” means a significant difference (*p* < 0.05), “**” means an extremely significant difference (*p* < 0.01), and “N.S.” indicates a non-significant difference.

**Figure 7 ijms-24-05663-f007:**
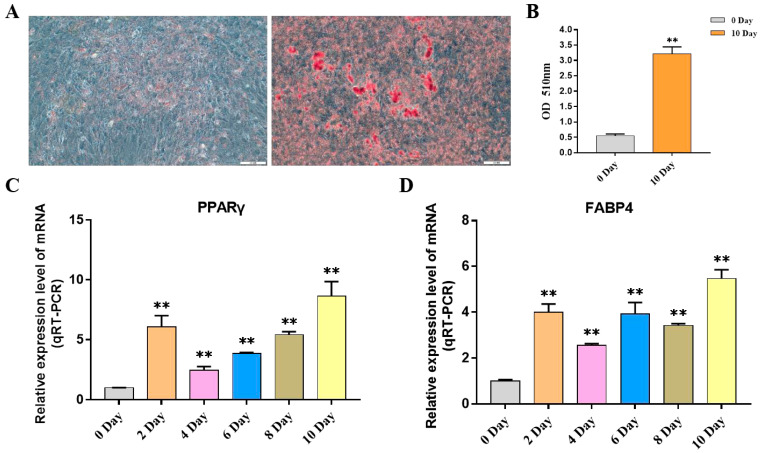
Induction of primary adipocyte differentiation. (**A**) Oil red O assay for lipid droplet distribution in adipocytes on day 0 and day 10 of differentiation. (**B**) The absorbance of adipocyte extract at 510 nm was measured at 0 and 10 days after induction. (**C**,**D**) The expression of *PPARγ* and *FABP4* genes was detected. Compared with the control, “**” means an extremely significant difference (*p* < 0.01).

**Figure 8 ijms-24-05663-f008:**
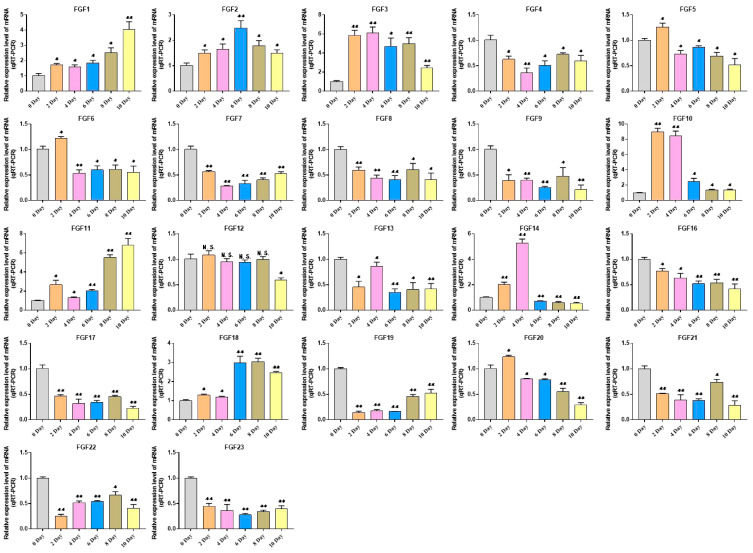
Detection of FGFs gene expression during adipocyte differentiation. Compared with the control, “*” means a significant difference (*p* < 0.05), “**” means an extremely significant difference (*p* < 0.01), and “N.S.” indicates a non-significant difference.

**Figure 9 ijms-24-05663-f009:**
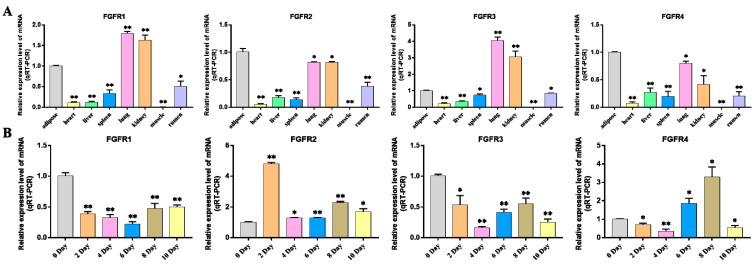
Expression assay of FGFRs genes. (**A**) Detection of FGFRs gene expression profile in different tissues of cattle. (**B**) Detection of FGFRs gene expression profile during bovine adipocyte differentiation. Compared with the control, “*” means a significant difference (*p* < 0.05), “**” means an extremely significant difference (*p* < 0.01).

**Figure 10 ijms-24-05663-f010:**
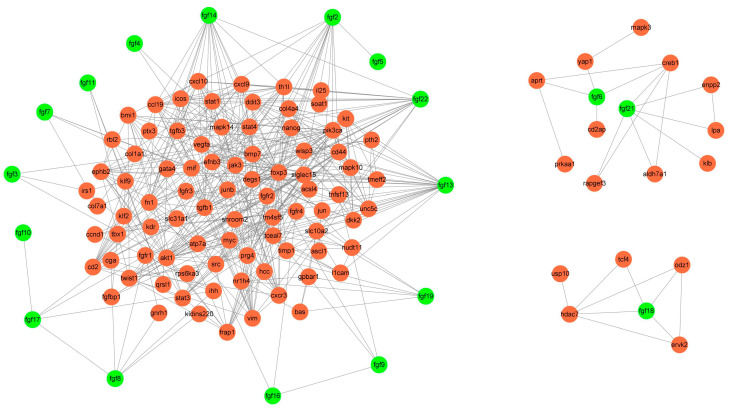
Interactive network mining of FGF genes.

**Table 1 ijms-24-05663-t001:** Description of *Bos taurus* FGF family genes.

Protein Name	Gene ID	Transcript ID	pI	Mw/Da	Amino Acids	Description
FGF1	ENSBTAG00000005198	ENSBTAT00000078959	6.51	17,492.86	156	Fibroblast growth factor 1
FGF2	ENSBTAG00000005691	ENSBTAT00000007477	9.58	17,249.83	156	Fibroblast growth factor 2
FGF3	ENSBTAG00000008623	ENSBTAT00000011373	11.44	26,672.65	237	Fibroblast growth factor 3
FGF4	ENSBTAG00000012563	ENSBTAT00000029946	9.89	22,068.57	207	Fibroblast growth factor 4
FGF5	ENSBTAG00000017348	ENSBTAT00000023064	10.67	29,640.87	271	Fibroblast growth factor 5
FGF6	ENSBTAG00000006800	ENSBTAT00000016675	10.24	22,623.16	209	Fibroblast growth factor 6
FGF7	ENSBTAG00000051898	ENSBTAT00000073054	9.35	22,489.17	195	Fibroblast growth factor 7
FGF8	ENSBTAG00000001530	ENSBTAT00000002001	10.36	27,605.6	245	Fibroblast growth factor 8
FGF9	ENSBTAG00000048237	ENSBTAT00000064356	7.06	23,382.45	209	Fibroblast growth factor 9
FGF10	ENSBTAG00000051910	ENSBTAT00000086367	9.61	23,768.15	214	Fibroblast growth factor 10
FGF11	ENSBTAG00000019242	ENSBTAT00000025622	10.06	25,081.1	226	Fibroblast growth factor 11
FGF12	ENSBTAG00000012413	ENSBTAT00000083812	9.28	23,236.66	208	Fibroblast growth factor 12
FGF13	ENSBTAG00000051480	ENSBTAT00000075597	9.15	28,757.89	256	Fibroblast growth factor 13
FGF14	ENSBTAG00000052496	ENSBTAT00000070063	9.32	20,378.4	185	Fibroblast growth factor 14
FGF16	ENSBTAG00000006722	ENSBTAT00000008839	9.36	23,791.89	208	Fibroblast growth factor 16
FGF17	ENSBTAG00000046951	ENSBTAT00000065993	10.5	24,919.41	217	Fibroblast growth factor 17
FGF18	ENSBTAG00000000128	ENSBTAT00000000139	9.86	23,920.57	208	Fibroblast growth factor 18
FGF19	ENSBTAG00000017285	ENSBTAT00000022973	7.7	24,072.59	219	Fibroblast growth factor 19
FGF20	ENSBTAG00000044043	ENSBTAT00000061318	8.54	23,489.59	212	Fibroblast growth factor 20
FGF21	ENSBTAG00000011624	ENSBTAT00000015438	6.08	22,585.83	210	Fibroblast growth factor 21
FGF22	ENSBTAG00000027357	ENSBTAT00000086016	11.48	23,305.82	219	Fibroblast growth factor 22
FGF23	ENSBTAG00000030343	ENSBTAT00000008940	9.64	26,733.24	246	Fibroblast growth factor 23

**Table 2 ijms-24-05663-t002:** Homology analysis of FGF gene between *Bos taurus* and other bovine subfamily species.

Gene	*Bos indicus*	*Hybrid-Bos taurus*	*Hybrid-Bos indicus*	*Bos grunnines*	*Bubalus bubalis*
*FGF1*	Y	Y	Y	Y	Y
*FGF2*	Y	Y	Y	Y	Y
*FGF3*	-	Y	Y	Y	Y
*FGF4*	Y	Y	Y	Y	Y
*FGF5*	-	Y	Y	Y	Y
*FGF6*	Y	Y	Y	Y	Y
*FGF7*	Y	Y	Y	Y	Y
*FGF8*	Y	Y	Y	Y	Y
*FGF9*	Y	Y	Y	Y	N
*FGF10*	Y	Y	Y	Y	Y
*FGF11*	N	N	N	N	N
*FGF12*	Y	Y	Y	Y	Y
*FGF13*	Y	-	-	Y	Y
*FGF14*	Y	Y	Y	Y	-
*FGF16*	Y	-	Y	Y	Y
*FGF17*	-	Y	Y	-	Y
*FGF18*	Y	Y	Y	Y	Y
*FGF19*	Y	Y	Y	-	Y
*FGF20*	Y	Y	Y	Y	Y
*FGF21*	Y	Y	Y	Y	Y
*FGF22*	Y	Y	Y	Y	Y
*FGF23*	Y	Y	Y	Y	Y

‘Y’ represents the synteny of genes between two species, while ‘N’ means not, and ‘-’ means lacking the gene.

## Data Availability

All data are reported in this manuscript.

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
