# Peer review of "Genome-Wide Identification and Characterization of Bovine Fibroblast Growth Factor (FGF) Gene and Its Expression during Adipocyte Differentiation"

_ijms, 2023, doi:10.3390/ijms24065663_

Round 1
Reviewer 1 Report
This manuscript attempts to identify the bovine FGF family and to perform a comprehensive analysis of their amino acid sequences, including phylogenetic trees and genetic structures such as genomic locations. This is considered interesting data, as it supports the subject includes almost all existing FGF families. However, there are several problems that listed below.
The main weakness of this manuscript is that this study seems to only reveal that FGFs present in other species were also present in cattle. I think that the originality and novelty of this study needs to be more clearly stated in this manuscript. Therefore, I recommend the paper require revision for publication in IJMS.
The following are my comments and suggestions:
1) The relevance of the data from the comprehensive analysis of bovine FGF and the results on adipocyte differentiation is tenuous. They do not seem to provide findings that are novel enough to be specified in the title of this manuscript, at least compared to results obtained in other species for adipocyte differentiation.
(2) There are not many differences on bovine FGF from existing findings about FGFs, making it difficult to judge the novelty of this study to be sufficient for publication in the IJMS. In Discussion part, the authors' thoughts on the novelty of their main focus in this manuscript should be clearly stated in detail.
3) There is no discussion in terms of comparisons between bovine FGF and other species, especially with rodents. At first glance, it appears that there are no significant differences in protein structure, tissue distribution and cellular involvement between cattle and other species. Consideration of this point is necessary in Discussion part.
Author Response
Dear Editors and Reviewers:
Firstly, we would like to thank you for your kind letter and for reviewers’ constructive comments concerning our manuscript entitled “Genome-wide identification and characterization of bovine Fibroblast growth factor (FGF) gene and its expression during adipocyte differentiation”. These comments are all valuable and helpful for improving our article. All the authors have seriously discussed about all these comments. According to the reviewers’ comments, we have tried best to modify our manuscript to meet with the requirements of your journal. In this revised version, changes to our manuscript within the document were all highlighted with tracked changes. Point-by-point responses to the reviewers are listed below.
Responds to the reviewer’s comments:
Reviewer #1:
This manuscript attempts to identify the bovine FGF family and to perform a comprehensive analysis of their amino acid sequences, including phylogenetic trees and genetic structures such as genomic locations. This is considered interesting data, as it supports the subject includes almost all existing FGF families. However, there are several problems that listed below.
The main weakness of this manuscript is that this study seems to only reveal that FGFs present in other species were also present in cattle. I think that the originality and novelty of this study needs to be more clearly stated in this manuscript. Therefore, I recommend the paper require revision for publication in IJMS.
Response: We feel great thanks for your professional work on our article. As you are concerned, there are several problems that need to be addressed. According to your nice suggestions, we have made extensive corrections to our previous manuscript, the detailed corrections are listed below.
The following are my comments and suggestions:
(1) The relevance of the data from the comprehensive analysis of bovine FGF and the results on adipocyte differentiation is tenuous. They do not seem to provide findings that are novel enough to be specified in the title of this manuscript, at least compared to results obtained in other species for adipocyte differentiation.
Response: Thank you very much for the suggestion. Gene family analysis can quickly identify the members of gene family related to target traits in the target species, and understand the evolution history of genes, which is helpful to narrow the scope of research and provide reference for subsequent genetic study. FGF genes play important role in regulating cell proliferation and differentiation, nutritional metabolism and neural activity. At present, the whole genome identification and analysis of bovine FGF gene family has not been carried out. Based on genomic data and annotation information, this study provided the first comprehensive analysis of the molecular characteristics, gene structure, chromosomal position, conserved motifs, and evolutionary relationships of FGF genes in bovine species using a bioinformatics approach. Furthermore, FGFRs are specific binding receptors for FGFs genes and most FGFs mediate their biological responses by binding and activating FGFRs. Therefore, we performed expression analysis of FGFs and FGFRs and the results showed that FGFs-FGFRs signalling might play an important role in the regulation of adipocyte differentiation (Figure 7, Figure 8, Additional file 6). As a reference, the results of this study laid a foundation for further exploration of the characteristics of FGF gene family, and also laid a foundation to further analyze the molecular mechanism of FGF1,FGF5,FGF10,FGF12,FGF16,FGF17 and FGF20 genes in the regulation of bovine adipogenic differentiation. In addition, based on your suggestion, we re-discuss the relationship between FGF gene and adipocyte differentiation, as follows (Lines 286 to 333 of the manuscript).
In order to understand the expression of FGF family members, the expression profiles of FGF gene in 8 tissue types of cattle were analyzed. The results showed that FGF family members were expressed in all these tissues, with FGF1, FGF5, FGF10, FGF12, FGF16, FGF17 and FGF20 being highly expressed in adipose tissue. Studies have shown that FGF1 can promote the differentiation of human preadipocytes into mature adipocytes by regulating the dependent network of BMP protein and activin membrane binding inhibitor (BAMBI)/PPARγ [31]. FGF2 can activate PI3K/AKT signal pathway and promote the proliferation and adipogenic differentiation of adipose stem cells [32]. FGF10 can promote adipogenic differentiation of goat intramuscular preadipocytes [33]. This is consistent with our qRT-PCR results, the expression levels of FGF1, FGF2 and FGF10 are significantly increased in induced adipocytes, and other FGF family members have different expression levels. Fibroblast growth factor receptors (FGFRs) are tyrosine kinase receptors (TRKs) that include four genes, FGFR1, FGFR2, FGFR3 and FGFR4 [34]. Four FGFR genes in vertebrates produce seven FGFR proteins with different ligand binding specificity (FGFRs1b,1c,2b,2c,3b,3c and 4) according to the difference of immunoglobulin-like domain III, and the biological function of typical FGFs is mediated by the interaction with FGFRs [35-37]. Therefore, we performed an expression analysis of FGFRs, which showed that FGFR2 and FGFR4 were most highly expressed in adipose tissue, and FGFR1 and FGFR3 were less expressed in adipose tissue than in lung and kidney tissue, but higher than in other tissues (Additional file 6). Previous studies on the binding specificity of FGFs-FGFRs found that members of the FGF7 subfamily were able to strongly activate FGFR2b, members of the FGF8 and FGF9 subfamilies showed high relative activity towards FGFR3c, members of the FGF19 subfamily showed consistent activity towards FGFR1c, 2c, 3c and FGFR4, and members of the FGF4 subfamily specifically activated the c receptor splice form [36, 37]. Our analysis of the expression of FGFRs during adipocyte lipogenic differentiation also showed that the expression trends of FGFRs were similar to their specifically bound FGF subfamily members, and the findings strongly suggest that FGFs-FGFRs signalling plays an important role in adipose tissue development and adipocyte lipogenic differentiation (Figure 7, Figure 8, Additional file 6).
The interaction between proteins can reveal their regulatory relationship, which helps us to understand the potential function of these proteins. We use Cytoscape's Agilent plug-in to mine the literature about FGF family members and their interaction genes to build a complete interaction network (Figure 9) [38]. For example, FGF signal can regulate the metabolism of endothelial cells through MYC-dependent HK2 expression, which in turn affects the development of blood vessels and lymphatic vessels [39]. Knockout of FGF21 gene in liver tissue can activate glucose-6-phosphatase and phosphoenolpyruvate carboxykinase through STAT3/SOCS3 pathway, thus increasing gluconeogenesis and glycogen decomposition, resulting in the aggravation of liver insulin resistance [40]. In addition, studies correlating single nucleotide polymorphisms (SNPs) in the 3' untranslated region (UTR) of the FGF21 gene with metabolic syndrome, obesity and diabetes showed that genetic variants in the 3' UTR region of the FGF21 gene were associated with obesity and not with metabolic syndrome and diabetes [41]. The opening of Piezo1 ion channels in mature adipocytes leads to the release of FGF1, which activates FGFR1 and induces precursor adipocyte differentiation [42]. In general, these results showed that the FGF gene family plays a role in regulating vascular development, metabolism and adipose differentiation by interacting with other genes.
References:
- He J, Chen DL, Samocha-Bonet D, Gillinder KR, Barclay JL, Magor GW, Perkins AC, Greenfield JR, Yang G, Whitehead JP. Fibroblast growth factor-1 (FGF-1) promotes adipogenesis by downregulation of carboxypeptidase A4 (CPA4) - a negative regulator of adipogenesis implicated in the modulation of local and systemic insulin sensitivity. Growth Factors. 2016 Dec;34(5-6):210-216. doi: 10.1080/08977194.2017.1285764.
- Lu GM, Rong YX, Liang ZJ, Hunag DL, Wu FX, Ma YF, Luo ZZ, Liu XH, Mo S, Li HM. FGF2-induced PI3K/Akt signaling evokes greater proliferation and adipogenic differentiation of human adipose stem cells from breast than from abdomen or thigh. Aging (Albany NY). 2020 Jul 24;12(14):14830-14848. doi: 10.18632/aging.103547.
- Xu Q, Lin S, Wang Y, Zhu J, Lin Y. Fibroblast growth factor 10 (FGF10) promotes the adipogenesis of intramuscular preadipocytes in goat. Mol Biol Rep. 2018 Dec;45(6):1881-1888. doi: 10.1007/s11033-018-4334-1.
- Price CA. Mechanisms of fibroblast growth factor signaling in the ovarian follicle. J Endocrinol. 2016 Feb;228(2):R31-43. doi: 10.1530/JOE-15-0414.
- Zhang K, Ealy AD. Disruption of fibroblast growth factor receptor signaling in bovine cumulus-oocyte complexes during in vitro maturation reduces subsequent embryonic development. Domest Anim Endocrinol. 2012 May;42(4):230-8. doi: 10.1016/j.domaniend.2011.12.006.
- Ornitz DM, Xu J, Colvin JS, McEwen DG, MacArthur CA, Coulier F, Gao G, Goldfarb M. Receptor specificity of the fibroblast growth factor family. J Biol Chem. 1996 Jun 21;271(25):15292-7. doi: 10.1074/jbc.271.25.15292.
- Zhang X, Ibrahimi OA, Olsen SK, Umemori H, Mohammadi M, Ornitz DM. Receptor specificity of the fibroblast growth factor family. The complete mammalian FGF family. J Biol Chem. 2006 Jun 9;281(23):15694-700. doi: 10.1074/jbc.M601252200.
- Shannon P, Markiel A, Ozier O, Baliga NS, Wang JT, Ramage D, Amin N, Schwikowski B, Ideker T. Cytoscape: a software environment for integrated models of biomolecular interaction networks. Genome Res. 2003 Nov;13(11):2498-504. doi: 10.1101/gr.1239303.
- Yu P, Wilhelm K, Dubrac A, Tung JK, Alves TC, Fang JS, Xie Y, Zhu J, Chen Z, De Smet F, Zhang J, Jin SW, Sun L, Sun H, Kibbey RG, Hirschi KK, Hay N, Carmeliet P, Chittenden TW, Eichmann A, Potente M, Simons M. FGF-dependent metabolic control of vascular development. Nature. 2017 May 11;545(7653):224-228. doi: 10.1038/nature22322.
- Wang C, Dai J, Yang M, Deng G, Xu S, Jia Y, Boden G, Ma ZA, Yang G, Li L. Silencing of FGF-21 expression promotes hepatic gluconeogenesis and glycogenolysis by regulation of the STAT3-SOCS3 signal. FEBS J. 2014 May;281(9):2136-47. doi: 10.1111/febs.12767.
- Zhang M, Zeng L, Wang YJ, An ZM, Ying BW. Associations of fibroblast growth factor 21 gene 3' untranslated region single-nucleotide polymorphisms with metabolic syndrome, obesity, and diabetes in a Han Chinese population. DNA Cell Biol. 2012 Apr;31(4):547-52. doi: 10.1089/dna.2011.1302.
- Wang S, Cao S, Arhatte M, Li D, Shi Y, Kurz S, Hu J, Wang L, Shao J, Atzberger A, Wang Z, Wang C, Zang W, Fleming I, Wettschureck N, Honoré E, Offermanns S. Adipocyte Piezo1 mediates obesogenic adipogenesis through the FGF1/FGFR1 signaling pathway in mice. Nat Commun. 2020 May 8;11(1):2303. doi: 10.1038/s41467-020-16026-w.
(2) There are not many differences on bovine FGF from existing findings about FGFs, making it difficult to judge the novelty of this study to be sufficient for publication in the IJMS. In Discussion part, the authors' thoughts on the novelty of their main focus in this manuscript should be clearly stated in detail.
Response: Thank you very much for the suggestion. Identifying gene families based on genome-wide data is of great significance for understanding gene evolution history, gene function and species differentiation. FGF genes have a variety of biological activities, including roles in angiogenesis, mitosis, cell differentiation, cell migration and tissue damage repair. So far, some members of the FGF family have been identified and analyzed in animal species, such as mice, humans, carp, Caenorhabditiselegans and zebrafish. However, the identification and functional study of bovine FGF gene family have not been reported. In recent years, the whole genome sequences of several bovine subfamily species, including Bos taurus, Bubalus bubalis, Bos mutus, Bison bison bison and Bos taurus taurus, have been published or updated, which provides an important basis for further study of the whole genome identification of the FGF family in bovine subfamily species. In view of the potential role of members of the FGF family, this study comprehensively identified and analyzed the FGF gene family in cattle for the first time, in order to better understand the characteristics of the members and structure of the family and provide new ideas for further functional analysis.
Based on your suggestion, we reanalyzed the differences in the identification of fibroblast growth factor gene families between bovine and other species in the discussion section, as follows (Lines 230 to 248 of the manuscript).
Studies have shown that the FGF family has experienced at least two major extensions: the first expansion increased the number of FGF from one or more primitive FGF genes to eight primitive FGF genes, forming the prototype of eight subfamilies. The second amplification occurs in the process of allelic evolution, which is mainly caused by genome replication [23]. In this study, a phylogenetic tree was constructed using 22 bovine fibroblast growth factor protein sequences and found that they could be clustered into eight subfamilies (FGF1, FGF3, FGF4, FGF7, FGF8, FGF9, FGF11 and FGF19 subfamilies), while the 22 FGF genes in the human and mouse genomes were clustered into seven subfamilies, with FGF3 identified as a member of the FGF7 subfamily [17, 21]. In all the current studies, FGF3 genes in vertebrates are always divided into FGF4 subfamily or FGF7 subfamily, but in fact, the clustering classification of FGF3 is still controversial [17, 23, 24]. Silvan proposes a new evolutionary scenario for FGF genes based on the results obtained by studying gene content, phylogenetic distribution and conservation of commonalities between amphioxus and vertebrates, namely that FGF3 forms a new subfamily, which is consistent with the results of the present study [25]. This evolutionary scheme is demonstrated for the first time in this study based on the results of analyzing the phylogenetic relationships of bovine FGF family members, which contributes to reconcile different evolutionary hypotheses proposed in previous studies.
References:
23.opovici C, Roubin R, Coulier F, Birnbaum D. An evolutionary history of the FGF superfamily. Bioessays. 2005 Aug;27(8):849-57. doi: 10.1002/bies.20261.
24.Coulier F, Pontarotti P, Roubin R, Hartung H, Goldfarb M, Birnbaum D. Of worms and men: an evolutionary perspective on the fibroblast growth factor (FGF) and FGF receptor families. J Mol Evol. 1997 Jan;44(1):43-56. doi: 10.1007/pl00006120.
25.Bertrand S, Camasses A, Somorjai I, Belgacem MR, Chabrol O, Escande ML, Pontarotti P, Escriva H. Amphioxus FGF signaling predicts the acquisition of vertebrate morphological traits. Proc Natl Acad Sci U S A. 2011 May 31;108(22):9160-5. doi: 10.1073/pnas.1014235108. Epub 2011 May 12.
(3) There is no discussion in terms of comparisons between bovine FGF and other species, especially with rodents. At first glance, it appears that there are no significant differences in protein structure, tissue distribution and cellular involvement between cattle and other species. Consideration of this point is necessary in Discussion part.
Response: Thank you very much for the suggestion. In the discussion part, we re-analyzed the differences between the identification of FGF gene family in Bovidae and other species, as follows (Lines 218 to 226, 230 to 248 and 271 to 281 of the manuscript.).
Previous studies have found that Caenorhabditis elegans has only two FGF genes, whereas 22, 22, 27 and 35 FGF genes were identified in the genomes of human, mouse, zebrafish and common carp, respectively, indicating that the massive expansion of FGF gene family members occurred during the evolution of primitive metazoans into vertebrates and aquatic organisms [17-21]. In addition, a new FGF gene, FGF24, has been identified in zebrafish, but direct homologues of FGF24 have not been identified in humans, mice or bovids, and it is speculated that it may have been lost during the evolution of these animals [17, 22].
Studies have shown that the FGF family has experienced at least two major extensions: the first expansion increased the number of FGF from one or more primitive FGF genes to eight primitive FGF genes, forming the prototype of eight subfamilies. The second amplification occurs in the process of allelic evolution, which is mainly caused by genome replication [23]. In this study, a phylogenetic tree was constructed using 22 bovine fibroblast growth factor protein sequences and found that they could be clustered into eight subfamilies (FGF1, FGF3, FGF4, FGF7, FGF8, FGF9, FGF11 and FGF19 subfamilies), while the 22 FGF genes in the human and mouse genomes were clustered into seven subfamilies, with FGF3 identified as a member of the FGF7 subfamily [17, 21]. In all the current studies, FGF3 genes in vertebrates are always divided into FGF4 subfamily or FGF7 subfamily, but in fact, the clustering classification of FGF3 is still controversial [17, 23, 24]. Silvan proposes a new evolutionary scenario for FGF genes based on the results obtained by studying gene content, phylogenetic distribution and conservation of commonalities between amphioxus and vertebrates, namely that FGF3 forms a new subfamily, which is consistent with the results of the present study [25]. This evolutionary scheme is demonstrated for the first time in this study based on the results of analyzing the phylogenetic relationships of bovine FGF family members, which contributes to reconcile different evolutionary hypotheses proposed in previous studies.
In the study of human FGF gene family, it was found that FGF3, FGF4 and FGF19 were located on chromosome 11, and the distances were 40kb and 10kb, respectively. FGF6 and FGF23 were within the 55kb on chromosome 12. In the mouse study, it was found that FGF3, FGF4 and FGF19 were located in the 80kb range of chromosome 7, and FGF6 and FGF23 were closely linked on chromosome 6 [28]. Like humans and mice, we identified FGF3, FGF4 and FGF19 on bovine chromosomes within the range of 35kb and 63kb on chromosome 29, while FGF6 and FGF23 were also located within 50kb on chromosome 5 (Figure 3). These gene positions suggest that the FGF gene family arose through duplication and translocation of genes and chromosomes during evolution, which would have contributed to the diversification of gene functions [28].
References:
- Itoh N, Ornitz DM. Evolution of the Fgf and Fgfr gene families. Trends Genet. 2004 Nov;20(11):563-9. doi: 10.1016/j.tig.2004.08.007.
- Itoh N, Ornitz DM. Functional evolutionary history of the mouse Fgf gene family. Dev Dyn. 2008 Jan;237(1):18-27. doi: 10.1002/dvdy.21388.
- Draper BW, Stock DW, Kimmel CB. Zebrafish fgf24 functions with fgf8 to promote posterior mesodermal development. Development. 2003 Oct;130(19):4639-54. doi: 10.1242/dev.00671.
- opovici C, Roubin R, Coulier F, Birnbaum D. An evolutionary history of the FGF superfamily. Bioessays. 2005 Aug;27(8):849-57. doi: 10.1002/bies.20261.
- Coulier F, Pontarotti P, Roubin R, Hartung H, Goldfarb M, Birnbaum D. Of worms and men: an evolutionary perspective on the fibroblast growth factor (FGF) and FGF receptor families. J Mol Evol. 1997 Jan;44(1):43-56. doi: 10.1007/pl00006120.
- Bertrand S, Camasses A, Somorjai I, Belgacem MR, Chabrol O, Escande ML, Pontarotti P, Escriva H. Amphioxus FGF signaling predicts the acquisition of vertebrate morphological traits. Proc Natl Acad Sci U S A. 2011 May 31;108(22):9160-5. doi: 10.1073/pnas.1014235108.
- Ornitz DM, Itoh N. Fibroblast growth factors. Genome Biol. 2001;2(3):REVIEWS3005. doi: 10.1186/gb-2001-2-3-reviews3005.
Reviewer 2 Report
This study identified the genomic distribution and the expression profile of bovine FGFs genes. Although detailed bioinformatics analyses were performed, the presentation was descriptive. The impact of the study was rather weak and the manuscript lacks a strong message for readers. Before publication in IJMS, the study should be deepen at least from the following standpoint.
Major concerns:
Not only the expression profile of FGFs, that of their receptors (FGFR1-4) is important to investigate the role for FGF signals. Similar analyses regarding bovine FGFRs along with a study about FGF-FGFR specificity (ref. Zhang X et al., J Biol Chem. 281: 15694–15700, 2006) should be performed.
Author Response
Dear Editors and Reviewers:
Firstly, we would like to thank you for your kind letter and for reviewers’ constructive comments concerning our manuscript entitled “Genome-wide identification and characterization of bovine Fibroblast growth factor (FGF) gene and its expression during adipocyte differentiation”. These comments are all valuable and helpful for improving our article. All the authors have seriously discussed about all these comments. According to the reviewers’ comments, we have tried best to modify our manuscript to meet with the requirements of your journal. In this revised version, changes to our manuscript within the document were all highlighted with tracked changes. Point-by-point responses to the reviewers are listed below.
Responds to the reviewer’s comments:
Reviewer #2:
This study identified the genomic distribution and the expression profile of bovine FGFs genes. Although detailed bioinformatics analyses were performed, the presentation was descriptive. The impact of the study was rather weak and the manuscript lacks a strong message for readers. Before publication in IJMS, the study should be deepen at least from the following standpoint.
Response: We feel great thanks for your professional review work on our article. As you are concerned, there are several problems that need to be addressed. According to your nice suggestions, we have made extensive corrections to our previous manuscript, the detailed corrections are listed below.
Major concerns:
Not only the expression profile of FGFs, that of their receptors (FGFR1-4) is important to investigate the role for FGF signals. Similar analyses regarding bovine FGFRs along with a study about FGF-FGFR specificity (ref. Zhang X et al., J Biol Chem. 281: 15694–15700, 2006) should be performed.
Response: Thank you very much for the suggestion. Based on your suggestion, we have supplemented the expression analysis of FGFR1-4, and the results are added to supplementary document 6. Meanwhile, we reanalyzed the specificity studies of FGFs with FGFRs in the Discussion section, as follows (Lines 297 to 315 of the manuscript).
Fibroblast growth factor receptors (FGFRs) are tyrosine kinase receptors (TRKs) that include four genes, FGFR1, FGFR2, FGFR3 and FGFR4 [32]. Four FGFR genes in vertebrates produce seven FGFR proteins with different ligand binding specificity (FGFRs1b,1c,2b,2c,3b,3c and 4) according to the difference of immunoglobulin-like domain III, and the biological function of typical FGFs is mediated by the interaction with FGFRs [33-35]. Therefore, we performed an expression analysis of FGFRs, which showed that FGFR2 and FGFR4 were most highly expressed in adipose tissue, and FGFR1 and FGFR3 were less expressed in adipose tissue than in lung and kidney tissue, but higher than in other tissues (Additional file 6). Previous studies on the binding specificity of FGFs-FGFRs found that members of the FGF7 subfamily were able to strongly activate FGFR2b, members of the FGF8 and FGF9 subfamilies showed high relative activity towards FGFR3c, members of the FGF19 subfamily showed consistent activity towards FGFR1c, 2c, 3c and FGFR4, and members of the FGF4 subfamily specifically activated the c receptor splice form [34, 35]. Our analysis of the expression of FGFRs during adipocyte lipogenic differentiation also showed that the expression trends of FGFRs were similar to the expression trends of their specifically bound FGF subfamily members, and the findings strongly suggest that FGFs-FGFRs signalling plays an important role in adipose tissue development and adipocyte lipogenic differentiation (Figure 7, Figure 8, Additional file 6).
References:
- Price CA. Mechanisms of fibroblast growth factor signaling in the ovarian follicle. J Endocrinol. 2016 Feb;228(2):R31-43. doi: 10.1530/JOE-15-0414.
- Zhang K, Ealy AD. Disruption of fibroblast growth factor receptor signaling in bovine cumulus-oocyte complexes during in vitro maturation reduces subsequent embryonic development. Domest Anim Endocrinol. 2012 May;42(4):230-8. doi: 10.1016/j.domaniend.2011.12.006.
- Ornitz DM, Xu J, Colvin JS, McEwen DG, MacArthur CA, Coulier F, Gao G, Goldfarb M. Receptor specificity of the fibroblast growth factor family. J Biol Chem. 1996 Jun 21;271(25):15292-7. doi: 10.1074/jbc.271.25.15292.
- Zhang X, Ibrahimi OA, Olsen SK, Umemori H, Mohammadi M, Ornitz DM. Receptor specificity of the fibroblast growth factor family. The complete mammalian FGF family. J Biol Chem. 2006 Jun 9;281(23):15694-700. doi: 10.1074/jbc.M601252200.
Round 2
Reviewer 1 Report
Comments to an Editor:
After a careful review of this resubmitted manuscript, all my reviewer`s comments of the weakness in previous version are clearly improved in this version. So, I recommended this article to be published in the IJMS.
Minar point
The FGF receptor in Supplementary figure 6 could be shown in the main text as it is important information for effect of each FGF family.
Author Response
Reviewer #1: After a careful review of this resubmitted manuscript, all my reviewer`s comments of the weakness in previous version are clearly improved in this version. So, I recommended this article to be published in the IJMS.
Minar point
The FGF receptor in Supplementary figure 6 could be shown in the main text as it is important information for effect of each FGF family.
Response: We once more are thankful to the Reviewers for their feedback. As per your suggestion, we have transferred the content of Additional file 6 to the body of the manuscript (Figure 9). Thank you for all the work you have done to review this manuscript.
Reviewer 2 Report
In the revised manuscript, authors have sufficiently addressed the concerns raised by the reviewer. The current manuscript has been improved to warrant publication in IJMS.
Author Response
Reviewer #2: In the revised manuscript, authors have sufficiently addressed the concerns raised by the reviewer. The current manuscript has been improved to warrant publication in IJMS.
Response: Thanks for all the work you have done to review this manuscript.